# A Simple Route of Printing Explosive Crystalized Micro-Patterns by Using Direct Ink Writing

**DOI:** 10.3390/mi12020105

**Published:** 2021-01-21

**Authors:** Albertus Ivan Brilian, Veasna Soum, Sooyong Park, Soojin Lee, Jungwook Kim, Kuktae Kwon, Oh-Sun Kwon, Kwanwoo Shin

**Affiliations:** 1Department of Chemistry, Institute of Biological Interfaces, Sogang University, Seoul 04107, Korea; ivan@sogang.ac.kr (A.I.B.); qkrtndyd111@naver.com (S.P.); oskwon@sogang.ac.kr (O.-S.K.); 2Graduate School of Science, Royal University of Phnom Penh, Phnom Penh 12150, Cambodia; 3Department of Chemical and Biomolecular Engineering, Sogang University, Seoul 04107, Korea; 0120sweety@naver.com; 4Agency for Defense Development, Daejeon 34186, Korea; kwonkt@add.re.kr

**Keywords:** micro-pattern, all-liquid ink, direct ink writing, RDX crystal, surface treatment

## Abstract

The production of energetic crystalized micro-patterns by using one-step printing has become a recent trend in energetic materials engineering. We report a direct ink writing (DIW) approach in which micro-scale energetic composites composed of 1,3,5-trinitro-1,3,5-triazinane (RDX) crystals in selected ink formulations of a cellulose acetate butyrate (CAB) matrix are produced based on a direct phase transformation from organic, solvent-based, all-liquid ink. Using the formulated RDX ink and the DIW method, we printed crystalized RDX micro-patterns of various sizes and shapes on silicon wafers. The crystalized RDX micro-patterns contained single crystals on pristine Si wafers while the micro-patterns containing dendrite crystals were produced on UV-ozone (UVO)-treated Si wafers. The printing method and the formulated all-liquid ink make up a simple route for designing and printing energetic micro-patterns for micro-electromechanical systems.

## 1. Introduction

Recently, the interest in fabricating energetic materials in designed patterns has been significant [1,2,3,4,5,6]. In the development of smart weapons and related equipment, energetic materials are expected to be integrated on a smaller scale into systems [2,3]. The application of microcomposites of explosives is thought to provide a less expensive, safer approach to the miniaturization of energetic devices and the development of pyrotechnics, propellants, and explosives [7,8]. Furthermore, energetic materials that can be deposited with microscopic patterns on flat substrates are essential for the development of safe Si-based initiator chips that can be used for the timed detonation of safe arm and fire (SAF) devices [9,10].

The main functions of SAF devices are keeping the detonator safe when not in use, arming the device, and initiating the one primary explosive necessary to ignite the munition or the secondary explosive when desired [10]. A prototype SAF device that consisted of a Si-based safe initiator was reported by Pezous et al. [11]. The Si-based safe initiator layer contains a micro-actuator, which will be ignited to generate gases that move the screen into the armed position, and a micro-initiator, which will ignite the primary explosive located in its cavity. The charging method for the energetic material in a micro-actuator and a micro-initiator cavity in integrated micro-electro-mechanical system (MEMS) and SAF devices, however, must be precise, automated, and safe [12,13,14]. The traditional ways, such as mold pressing and cast molding, are inadequate for fusing in MEMSs due to the slow solidification of the drug column and the lack of accuracy in forming a small-sized explosive column.

Many printing methods have been used to generate micro-patterns for the deposition of conductive, ferromagnetic, piezoelectric, and energetic materials [9,15,16,17]. Those functional patterns can be designed and printed directly from electronic designed files without the need for a mask or a mold. However, the printing of energetic materials from an all-liquid ink to create micro-patterns is generally difficult because the typical solvent used in the ink and the solid crystals can cause malfunctions in the printing head and clog the nozzle, respectively. That the low surface tension of the ink, which is caused by the solvent used in the ink, limits printability is another disadvantage.

In this work, we demonstrated a novel, one-step method to print 1,3,5-trinitro-1,3,5-triazinane (RDX) with microscopic patterns by using a direct ink writing (DIW) approach with all-liquid inks. DIW is commonly used in fabricating patterns and offers significant advantages: it can be used to deposit complex structures and to assemble materials with tailored architectures on millimeter and micrometer length scales [6,18]. In addition, all-liquid inks offer several advantages: no additional processing steps are needed to produce printable nanomaterials, and it is easy to use. Therefore, the goal of this study was to investigate the crystal morphology and the precision of patterns with RDX-based composite materials fabricated using DIW with all-liquid inks.

## 2. Materials and Methods

### 2.1. Chemicals and Materials

1,3,5-trinitro-1,3,5-triazinane (RDX) crystal powder with particle sizes of 107 μm to 391 μm was obtained from the Agency for Defense Development (ADD), Daejeon, South Korea. ɣ-butyrolactone (GBL) and cellulose acetate butyrate (CAB; average Mn: ~65,000) were purchased from Sigma-Aldrich (St. Louis, MO, USA). P-type (boron-doped) silicon wafers with thicknesses of 525 ± 25 µm and polished on a single side, which were purchased from Silicon Technology Co. (Tokyo, Japan), were used as the printing substrates.

### 2.2. Preparation of RDX Inks

Raw RDX crystal powder and CAB were dissolved in ɣ-butyrolactone (RDX-CAB-GBL) in a RDX:CAB ratio of 9:1. Similarly, RDX crystal powder without CAB was dissolved in GBL (RDX-GBL). The final concentration of the ink was 0.55 M for both RDX-CAB-GBL and RDX-GBL. After all the materials had been dissolved in the solvent by vortexing at 500 rpm overnight, the explosive ink was filtered using a 5-µm PTFE syringe filter (Whatman^TM^, GE Healthcare, Buckinghamshire, UK) to prevent any undissolved materials and/or other contaminants from being unwittingly included in the ink. The ink’s viscosity was measured by using a rheometer (RVDV-III Ultra, Brookfield, MA, USA).

### 2.3. Preparation of the Silicon-Wafer Substrates

The substrates that were used in the experiment were silicon wafers with one side polished. Prior to the surface cleaning treatment, the Si-wafers had been cut into smaller pieces of approximately 2 × 2 cm^2^. The silicon wafers were then cleaned with deionized water, sonicated, and dried in an oven. After all the water had been evaporated from the surfaces of the substrates, the cleaned silicon wafers were exposed to ultraviolet (UV) light in a UV-ozone (UVO) cleaner (AH-1700, Ahtech LTS, Busan, Korea) for 5 min to increase their hydrophilicity.

### 2.4. Printing of RDX Inks

A microplotter (Sonoplot GIX, Sonoplot Inc., Middleton, WI, USA) was used to print the explosive inks. The ink dispenser was composed of a hollow, tapered, glass needle attached to a piezoelectric element and was capable of creating fluid patterns as small as 5 µm wide on a surface (Figure 1). The procedure for printing fine line patterns of explosive ink was optimized, and after optimization, fine line patterns could be printed using a 260, 130, 60, and 40 μm nozzle, a 15 V nozzle strength, and a 10 mm/s^2^ acceleration. The baseline printing conditions were single-layer printing with a 5 mm length for every design and individual ink droplets spaced 500 µm apart in the dot/droplet patterns. The nozzle’s tip was placed 10 µm above the surface of the substrate under ambient conditions at room temperature. The ink deposited by using the direct ink writing method crystalized spontaneously at room temperature.

### 2.5. Characterization of the Printed RDX Patterns

The RDX crystals were observed using a stereomicroscope (JSZ-7XB/7XT, Samwon, Seoul, Korea). The crystal morphologies of the printed samples were obtained using field emission scanning electron microscopy (FE-SEM) (JSM-7100F, JEOL Ltd., Tokyo, Japan). X-ray diffraction (XRD) was used to characterize the crystal structure of the RDX in the printed samples. The printed RDX patterns were characterized using an UltimaIV X-ray diffractometer, and the drop-casted RDX was characterized using a Rigaku diffractometer. Silicon substrates were used for the XRD characterization and were placed in the equipment’s sample holder “as-printed”.

## 3. Results

### 3.1. Direct Ink Writing with RDX Ink and Its Printability

In piezoelectric devices, the volumes of the droplets from the nozzle are on the order of 10–100 pL, depending on the diameter of the cavity’s orifice, the characteristics of the piezoelectric driving waveform, the fluidic pressure, and the rheological properties of the fluid [19,20]. We adjusted the printing speed and acceleration to 0.5 mm/s and 10 mm/s^2^, respectively, to enhance printability.

To investigate the printability of RDX inks, we formulated two liquid RDX inks, RDX-GBL ink and RDX-CAB-GBL ink, and we printed those inks on pristine Si wafers. The RDX-GBL ink had a viscosity of 2.16 cP while the RDX-CAB-GBL ink had a viscosity of 16.75 cP. The viscosity was increased in the latter because a polymeric binder (CAB), the use of which resulted in improved printability, had been added in the ratio of 1:9 (CAB:RDX); the printed patterns are shown in Figure 2a. Even though the patterns had been printed using the same ink and printing setup, their widths showed large variations because of the low surface wettability of the surface. The crystals were aggregated in the RDX-CAB-GBL ink due to the presence of the binder whereas the RDX-GBL ink without a binder showed poor printability, with only a small number of crystals being deposited on the substrate because the amount of RDX precursor was less. The addition of the polymeric binder to the formulation increased the printability of the ink because it allowed the explosive crystals to bind together in a matrix [21,22,23]. During the printing of RDX patterns, the solvent continuously evaporated from the printed RDX patterns to form crystalized RDX patterns. The selected RDX ink for further printing was RDX-CAB-GBL ink with an RDX concentration of 0.55 M.

To study the effects of the DIW method on the crystallinity of the printed RDX crystals, we conducted XRD at diffraction angles ranging from 10° to 40°. In Figure 2b, the XRD data for the RDX crystals printed on the surfaces of the Si wafers by using composite inks are compared to the XRD data for raw RDX powder; similar diffraction peaks are found at diffraction angles between 20° and 30°. In contrast to the raw RDX, the peak intensities of the RDX crystals in the patterns printed using the DIW method are weaker. Such weak intensities are attributed to a reduction in the content of RDX in the DIW ink. The differences in XRD peak profiles indicate that variable crystallographic phases and shapes were obtained [12,24]. The measured sample was in the form of dendrite-like crystals, which was different from the single crystal form of raw RDX powder attributed to the peak shift.

### 3.2. The Effect of Surface Treatment on Printed RDX Micro-Patterns

The purpose of the treatment was to improve the wettability of the Si substrate and eventually the printability of the ink. The wetting property of a substrate, as expressed by the liquid-substrate contact angle, reflects the extent to which an ink drop will spread on the substrate [25]. After the UVO oxidation treatment, the Si wafer was converted into a highly hydrophilic surface with a water contact angle of 28° while the pristine Si wafer had a surface contact angle of 63°.

When RDX patterns were printed on pristine Si wafers at a printing speed of 0.5 mm/s (Figure 3a–c), the printed micro-patterns were observed to have mostly an interesting single crystal structure of several individual or assembled crystals with sizes ranging from 3 µm to 40 µm distributed along the patterned lines (Figure 3c,d). This occurred because the lower surface energy of the untreated Si wafer was unfavorable for spreading. Furthermore, when an RDX ink is used to print a pattern on the surface of a pristine Si wafer, obtaining a pattern with good resolution is somehow difficult, as shown in Figure 2a and Figure 3a,b. Moreover, the distribution of the single crystals is inhomogeneous; as a result, some patterns have a disconnected appearance. Because these issues are caused by the low surface energy of the pristine Si wafer, we used RDX ink to print patterns on UVO-treated Si wafers and observed the differences.

We printed RDX patterns on the UVO-treated surfaces of the Si wafers, which are highly hydrophilic surfaces, and observed that the crystal morphology of the RDX particles was dendritic (Figure 4a,b). Long and smooth branches ranging in length from 70 µm to 122 µm were generated from the outer line of the RDX pattern toward the middle of the pattern, with most of the short branches being formed at the periphery (Figure 4b,c). Energy dispersive spectroscopy (EDS) confirmed the presence of elemental nitrogen from the RDX material in the crystalized printed patterns (Figure 4d,e). Based on these results, we infer that the orientation of the dendritic crystal structure formed on the UVO-treated surface is affected by the higher surface energy of that surface. Moreover, at a slower speed, more ink can be dispensed, allowing the growth of bigger crystals. In summary, printing RDX ink on the treated surface of a Si wafer provided better printability and allowed long and connected crystalized micro-patterns to be formed.

### 3.3. Printing of Long Micro-Pattern Arrays of RDX

We used RDX ink to fabricate long, dendrite-like crystalized RDX micro-patterns on UVO-treated surfaces of Si wafers, and we observed that the widths of those patterns depended on the printing speed and the nozzle size (Figure 5a,b). The width of the pattern could be reproduced with small variation when it was printed as a line and the printing speed was faster than 0.5 mm/s. Moreover, the printing of a non-linear pattern resulted in a large variation in the width of that pattern because, at the turning point, the speed of printing was relatively slow compared to the initial speed, which allowed the ink to be deposited in larger amounts.

Because the DIW system provides flexibility in designing and printing, various sizes and shapes of patterns can be printed without the need for a mask or a mold. This feature allows the quick, easy printing of crystalized RDX patterns for testing the performance of materials for use in MEMS. As a proof of concept, we designed and printed long, connected patterns on Si wafers with UVO-treated surfaces by using a 60-μm nozzle at a printing speed of 0.5 mm/s. Those patterns with dendrite-like RDX crystal structures had various shapes, such as dots, waves, alphabetic letters, and grids (Figure 5c–f). Even though we were able to generate long micro-patterns containing RDX crystals, we were unable to test the explosive behavior of those crystals due to safety regulations.

## 4. Conclusions

A DIW method using a microplotter was utilized for one-step, direct ink writing of an organic, solvent-based, all-liquid ink that contained an explosive RDX as its main component. The RDX ink at a concentration of 0.55 M, with cellulose acetate butyrate as a binder and ɣ-butyrolactone as its solvent, demonstrated superior printability. Micro-patterns of RDX single and dendrite crystals were fabricated on pristine and UVO-treated surfaces of Si wafers, respectively. Overall, the results suggest that the surface wettability significantly affected the printability and the morphology of the printed RDX crystals. Recently, novel MEMS-based SAF designs with increased reliability have been proposed, and we envision that the DIW method we reported here may provide an effective and reliable way to integrate energetic materials into new, complex, and compact SAF devices.

## Figures and Tables

**Figure 1 micromachines-12-00105-f001:**
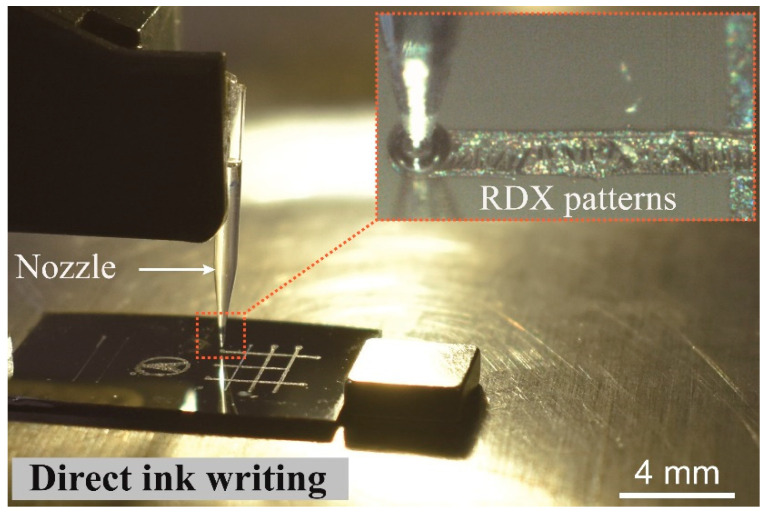
Picture of the direct ink writing system, with a focus on the dispensing unit (nozzle). The inset shows 1,3,5-trinitro-1,3,5-triazinane (RDX) ink being dispensed.

**Figure 2 micromachines-12-00105-f002:**
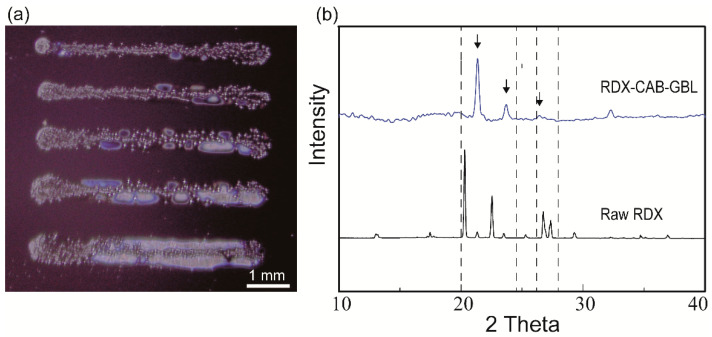
(**a**) Patterns containing RDX were printed on pristine Si wafers by using the formulated RDX-CAB-GBL ink. (**b**) XRD diffraction patterns of the different RDX crystal on Si substrates: raw RDX crystal powder and crystal RDX patterns. CAB: cellulose acetate butyrate; GBL: ɣ-butyrolactone.

**Figure 3 micromachines-12-00105-f003:**
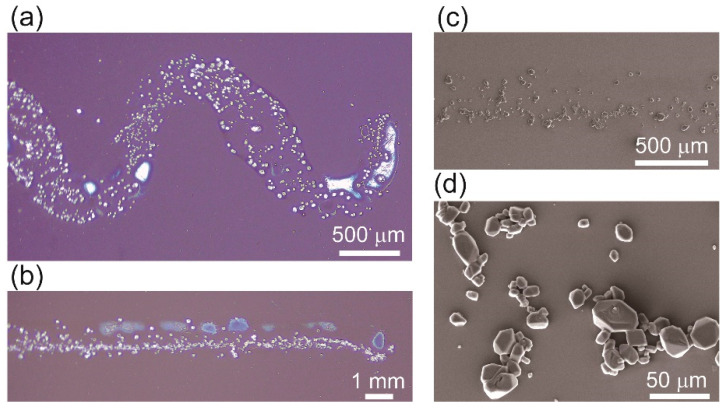
(**a**,**b**) RDX micro-patterns printed on a pristine Si-wafer surface. (**c**,**d**) SEM images of an RDX micro-pattern showing numbers of single RDX crystals in the pattern.

**Figure 4 micromachines-12-00105-f004:**
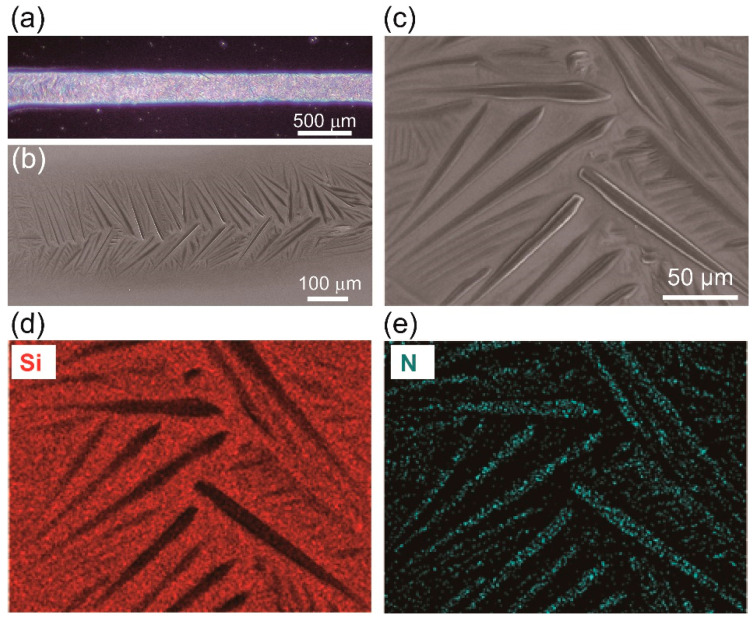
(**a**) An RDX micro-pattern printed on the UV-ozone (UVO)-treated surface of a Si wafer. (**b**) Dendrite crystal structures of RDX grown along the micro-pattern. (**c**) Higher magnification image of the dendrite crystal structure. (**d**,**e**) EDS composition mapping of the surface shown in (**c**).

**Figure 5 micromachines-12-00105-f005:**
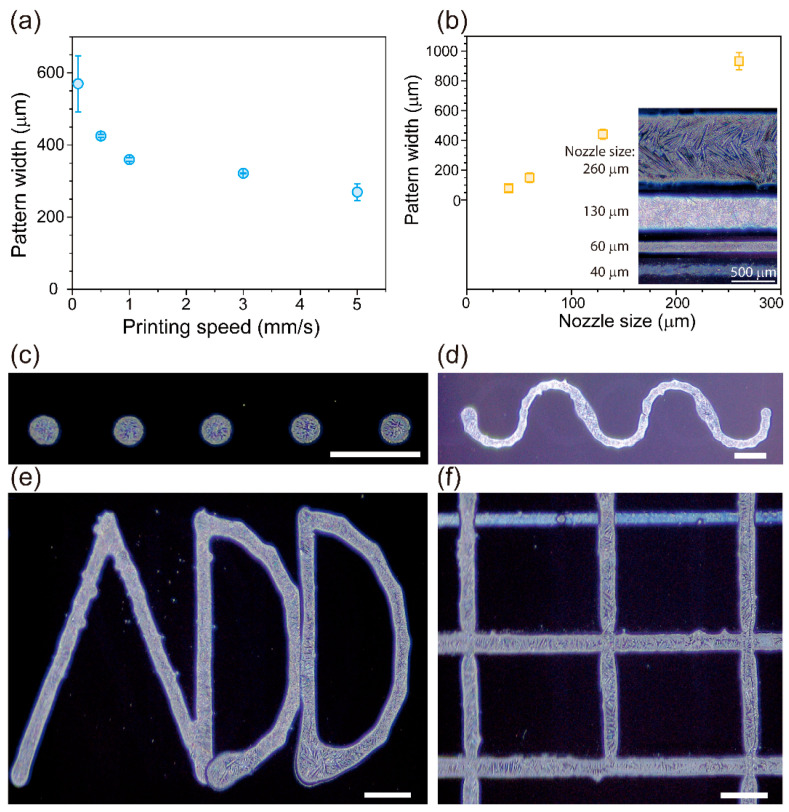
Widths of RDX patterns printed on the surfaces of UVO-treated Si wafers (**a**) at different speeds and (**b**) with different nozzle sizes (inner diameters). The inset of (**b**) shows representative RDX patterns printed by using various nozzle sizes from 40 μm to 260 μm. Various printed RDX patterns with different shapes were realized: (**c**) dot, (**d**) wavy, (**e**) alphabetic “ADD”, and (**f**) grid. They were printed on the UVO-treated surfaces of Si wafers. Scale bar = 500 μm.

## Data Availability

Data sharing not applicable.

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
