# Peer review of "A Simple Route of Printing Explosive Crystalized Micro-Patterns by Using Direct Ink Writing"

_micromachines, 2021, doi:10.3390/mi12020105_

Round 1

Reviewer 1 Report

1# For DIW, ink viscosity is an important parameter. It is better to provide the viscosity of ink used in this manuscript.

2# There are five printed lines in Figure 2a, what are the differences (different ink concentration?)? please mark.

3# “Even the RDX printed micro-patterns were observed mostly as a single crystal struc- ture, which is an interesting crystal structure, the printability of the RDX ink on the pris- tine Si-wafer surface is somehow difficult to maintain a good resolution of the printed pattern as have been shown in Figure 2a.” it should be in Figure 3a.

4# In Figure 5, if possible, please also provide the (optical) images of patterns demonstrated in Figure 5a and b.

Author Response

Response Letter to Reviewer 1#

We are naturally pleased with the response of the reviewers to our work on “A simple route of printing explosive crystalized micro-patterns by using direct ink writing”. We are particularly indebted to reviewers for the careful reading and insightful comments. Besides finding a number of errors, the comments prompted us to take a careful look at the details of our error estimations which has taken us some time to fix. Below we provide specific responses to the individual comments were made. We have also taken the opportunity to improve and add some sentences in the text, the captions, and the figure.

Yours sincerely,

Kwanwoo Shin

Department of Chemistry and Institute of Biological Interfaces

Sogang University, 35 Baekbeom-ro, Mapo, Seoul 04107, Korea

Tel.: +82-2-705-8441

Fax: +82-2-701-0967

Reviewer 2 Report

The paper presents an interesting approach for direct inkjet printing of explosive materials. To me it is surprising that the handling in the printing setup seems to be safe. There are some questions to the authors:

  1. It is not clear whether the crystals are dissolved and the ink formulation process or whether they are just dispersed. What is the pristine size of the crystal prior to the formulation? Could you comment on this? If the material dissolves in the solvent, how is the recrystallization process controlled? Can you comment on the drying process and how it influences the crystallization, crystal sizes and may be crystal structure.
  2. The XRD pattern of the prisitine material and the printed ones show differences in the peak pattern: peaks of the printed crystals are broader and shifted to higher Theta angles. Could you give an explanation for this observation?
  3. Figure 4 shows printed patterns on a treated surface, where obviously different crystal structures are observed than on untreated surfaces. Is this only cause by the better wetting of the solvent on the treated Si surface?

Author Response

Response Letter to Reviewer 2#

We are naturally pleased with the response of the reviewers to our work on “A simple route of printing explosive crystalized micro-patterns by using direct ink writing”. We are particularly indebted to reviewers for the careful reading and insightful comments. Besides finding number of errors, the comments prompted us to take a careful look at the details of our error estimations which has taken us some time to fix. Below we provide specific responses to the individual comments were made. We have also taken the opportunity to improve and add some sentences in the text.

Yours sincerely,

Kwanwoo Shin

Department of Chemistry and Institute of Biological Interfaces

Sogang University, 35 Baekbeom-ro, Mapo, Seoul 04107, Korea

Tel.: +82-2-705-8441

Fax: +82-2-701-0967
